# Household-level effects of seasonal malaria chemoprevention in the Gambia
Seyi Soremekun [1] ✉, Bakary Conteh[2], Abdoullah Nyassi[2], Harouna M. Soumare[2], Blessed Etoketim[2], Mamadou Ousmane Ndiath[2], John Bradley [3], Umberto D'Alessandro[2], Teun Bousema [4], Annette Erhart[2], Marta Moreno[1] & Chris Drakeley [1] ✉

## Abstract

**Background** In 2022 the WHO recommended the discretionary expansion of the eligible age range for seasonal malaria chemoprevention (SMC) to children older than 4 years. Older children are at lower risk of clinical disease and severe malaria so there has been uncertainty about the cost-benefit for national control programmes. However, emerging evidence from laboratory studies suggests protecting school-age children reduces the infectious reservoir for malaria and may significantly impact on transmission. This study aimed to assess whether these effects were detectable in the context of a routinely delivered SMC programme.

**Methods** In 2021 the Gambia extended the maximum eligible age for SMC from 4 to 9 years. We conducted a prospective population cohort study over the 2021 malaria transmission season covering 2210 inhabitants of 10 communities in the Upper River Region, and used a household-level mixed modelling approach to quantify impacts of SMC on malaria transmission.

**Results** We demonstrate that the hazard of clinical malaria in older participants aged 10+ years ineligible for SMC decreases by 20% for each additional SMC round per child 0–9 years in the same household. Older inhabitants also benefit from reduced risk of asymptomatic infections in high SMC coverage households. Spatial autoregression tests show impacts are highly localised, with no detectable spillover from nearby households.

**Conclusions** Evidence for the transmission-reducing effects of extended-age SMC from routine programmes implemented at scale has been previously limited. Here we demonstrate benefits to the entire household, indicating such programmes may be more cost-effective than previously estimated.

## Plain language summary

Seasonal malaria chemoprevention (SMC) is the provision of monthly, preventative, anti-malaria medication to young children at times when they are most at risk of severe disease. Recently the World Health Organisation recommended expanding SMC to children older than 4 years. Older children with malaria typically remain symptomless so the advantages were unclear. However, laboratory evidence suggests this group continues to transmit malaria to others. We conducted a population study in 2021 in 10 communities in the Gambia where SMC was extended to all children up to 9 years of age for the first time. We found household members aged over 9 years were less likely to get clinical disease when most young children in the same household did receive SMC. This suggests an added protection of SMC for those who do not receive it, potentially increasing cost-effectiveness.

Seasonal malaria chemoprevention (SMC) with sulphadoxine-pyrimethamine and amodiaquine (SP-AQ) is a World Health Organisation-endorsed strategy to prevent severe malaria and malaria deaths in young children in living in the African Sahel/sub-Sahel, where most transmission occurs within a few months of the year[1]. Since this recommendation, SMC has been implemented in more than 10 countries covering over 20 million children. Trials have shown that SMC reduces clinical malaria in children aged 0–4 years by more than 80% in the first 4 weeks following a dose, and 67% 4–6 weeks following a dose[2,3]. In

updated 2022 malaria control guidelines the WHO recommended control programmes could increase the SMC-eligible age to children above 4 years[4] to potentially mitigate a potential shift of the burden of severe malaria to older children as malaria transmission declines in endemic settings[5–7]. This was based on evidence from a 2010 stepped-wedge trial in Senegal showing a reduction of 61% in clinical cases in children 5–9 years receiving SMC[4,6]. However, more recent studies show mixed impact in this age group[8,9] and concerns about programme sustainability and maintenance of effective coverage in the current 0–4 age group have led

[1]Department of Infection Biology, London School of Hygiene & Tropical Medicine, Keppel Street, London, UK. [2]Medical Research Council Unit The Gambia at the London School of Hygiene & Tropical Medicine, Banjul, The Gambia. [3]Medical Research Council International Statistics and Epidemiology Group, London School of Hygiene & Tropical Medicine, Keppel Street, London, UK. [4]Department of Medical Microbiology, Radboud University Medical Center, Nijmegen, The Netherlands. ✉e-mail: Seyi.Soremekun@lshtm.ac.uk; Chris.Drakeley@lshtm.ac.uk

to reservations about the feasibility and resources required to implement an age increase[10].

Children aged over 5 years however have an increasing risk of asymptomatic infection and elevated gametocytaemia[11]; recent studies of controlled mosquito feeding experiments[12–15] and simulation analyses[16] have demonstrated asymptomatic infections in children in this age group are also often the largest contributors to malaria transmission. This suggests extended-age SMC, besides reducing clinical illness in direct recipients, may have wider impacts on transmission to non-recipients[6,17,18]. For policy-makers with finite resources, interventions that maximise impact on both disease transmission and clinical disease burden are prioritised, particularly when the goal is elimination[19]. The transmission-reducing impact of vector control interventions are well-described[20–22] and incorporated into models of cost-effectiveness by design as programme impact is frequently measured in the entire population targeted[20,23,24]. Vector control programmes are predicted to be highly cost-effective and usually prioritised for implementation[19,24]. The impacts of SMC programmes are normally measured only in eligible children, thus the true value of extended-age implementation is likely to be persistently underestimated, highlighting a crucial evidence gap for implementation design and policy.

In 2021, the National Malaria Control Programme (NMCP) of the Gambia Ministry of Health recommended increasing the eligible age for SMC to 9 years[25]. Given the highly clustered nature of malaria transmission, particularly in this context[26,27], herd impacts of the SMC programme should be primarily detectable within mixed-age households using household-level statistical approaches. Therefore as part of a larger study of malaria infection and transmission dynamics conducted in 2019–2022, 'The INDIE Trial'[28], we conducted a prospective cohort study collecting population data on SMC coverage and malaria burden across communities in Gambia's Upper River Region to assess the individual and household-level impacts of the extended programme. We find that in households where most children had received at least one round of SMC, the incidence of clinical malaria and prevalence of asymptomatic infections is lower in adolescents and adults who are themselves non-eligible for SMC compared to households with lower coverage of SMC in children—these results are robust to adjustment. This suggests transmission effects are detectable in routine programmes and extended-age SMC may have additional cost-benefits beyond reducing disease in target children.

## Methods
### Study site
The study was implemented in ten communities (Njayel, Banni Kunda, Temanto, Bolibana, Fula Mori Bochi, Madina Samba Sowe, Njum Bakary, Sare Demba Dardo, Sare Biram and Tabajang) in the Upper River Region (URR) of the Gambia (Fig. 1). The region has a population of over 250,000 inhabitants and has a distinctly seasonal pattern of malaria transmission, where the most cases occur between June and December, peaking directly following the annual rains between June and September[29]. The main vector species is *Anopheles gambiae sensu latu*[30,31]. During the 2021 season, the mean nightly *Anopheles gambiae s.l.* and *Anopheles funestus* catch rate per house in the study communities using Centres for Disease Control light traps rate was 0.97 (standard deviation 1.41) (Supplementary File STable 1). The peak temperature during the transmission season was 35.2 °C in 2021 (Supplementary File STable 2). A region-wide distribution drive of long lasting insecticide treated nets occurred in 2019. For the first time in July 2021, The NMCP extended the upper eligible age for SMC from 4 to 9 years. Eligible children aged 0–9 years could receive up to four rounds of monthly SMC with sulphadoxine-pyrimethamine (SP) and amodiaquine (AQ) administered between August and November 2021 by community outreach teams.

### Study design
This prospective observational cohort study was nested within a larger trial measuring the effectiveness of control interventions on malaria infection and transmission, the INDIE Trial (*P. falciparum* Infection Dynamics and

Transmission to Inform Elimination, clinicaltrials.gov reference NCT04053907) and used data collected between 26th July 2021 and 12th January 2022. Three communities (Njum Bakary, Sare Demba Dardo and Sare Biram) were randomly assigned to receive 3 monthly rounds of mass drug administration with dihydroartemisinin-piperaquine between April and June 2021. In two communities (Madina Samba Sowe and Tabajang) inhabitants were screened weekly for fever—'fever screen and treat (FSAT)', those with temperatures above 37.5 °C or a history of fever in the last week received a rapid diagnostic test (RDT) for malaria and positive cases were treated with artemether-lumefantrine. Three communities (Banni Kunda, Temanto and Njayel) were assigned to monthly screening of all residents with ultrasensitive-RDTs, and positive cases treated with artemether-lumefantrine—'mass screen and treat' (MSAT). The remaining two communities (Bolibana and Fula Mori Botchi) were assigned to the control group and received standard control interventions. A programme of community case management of malaria was concurrently initiated in all communities. From 26 July 2021, a clinician and support worker (nurse and community health worker) based in each community managed all suspected malaria cases (passive case detection—PCD).

### Study procedures
A series of community-based sensitisation activities took place during the INDIE baseline year (2019) to inform community members about the study and its aims, and all inhabitants were invited to take part. Residents who provided informed consent were enroled during a baseline dry season survey in April-May 2021. Cross-sectional surveys covering all enroled community members were carried out on a rolling basis every 8 weeks during the transmission season. Blood samples collected from the enroled population during all surveys were analysed at the MRC Unit The Gambia (MRCG) in Basse by quantitative polymerase chain reaction (qPCR varATS) to detect malaria infections. Study questionnaires were administered during the surveys to all participants and collected address details and GPS coordinates, demographics including age, gender and ethnicity, symptoms of illness, care seeking and any treatments received for each participant and use of insecticide-treated bed nets the night prior to the survey. Identical data were collected from malaria patients identified by PCD visits although diagnosis was done by RDT only. The dates and number of all SMC rounds received for each child under 10 years were collected from SMC cards or by caregiver report. SMC data were collected in a survey after the last (November) round of SMC, and at two additional capture points: during a final cross-sectional survey in January 2022 and following a review of the SMC database by SS and AN, in a dedicated SMC mop-up survey to address data queries and collect data on additional study children not met in previous surveys (10th–15th March 2022). Entomological surveillance was conducted during the 2021 study period in 6 randomly selected households per community. In each household, Centre for Disease Control (CDC) light traps were hung in a sleeping room for three consecutive nights per month; caught mosquitoes were examined to determine species, sex and gonotrophic status (fed status and gravidity).

### Study outcomes
All data from PCD visits, cross-sectional and entomological surveys were collected onto secure handheld devices and stored, cleaned and accessed via a password protected REDcap® server (Vanderbilt University, Nashville Tennessee) hosted at the MRC LSHTM Research Unit in Fajara. Incidence of clinical malaria was defined as the number of passively detected cases per 100 person-months. For each clinical case, 2 weeks of follow up were removed from the denominator. Malaria prevalence was defined as the percentage of participants sampled who were qtPCR-positive during a late season survey (27th September to the 27th November 2021). qtPCR-positive participants sampled during this survey were categorised into high- and low-density infections by age group. High-density infections were those with density above the median of the natural log asexual parasite concentrations per µL blood for the age groups (0–4, 5–9, 10–15 and 16+ years).

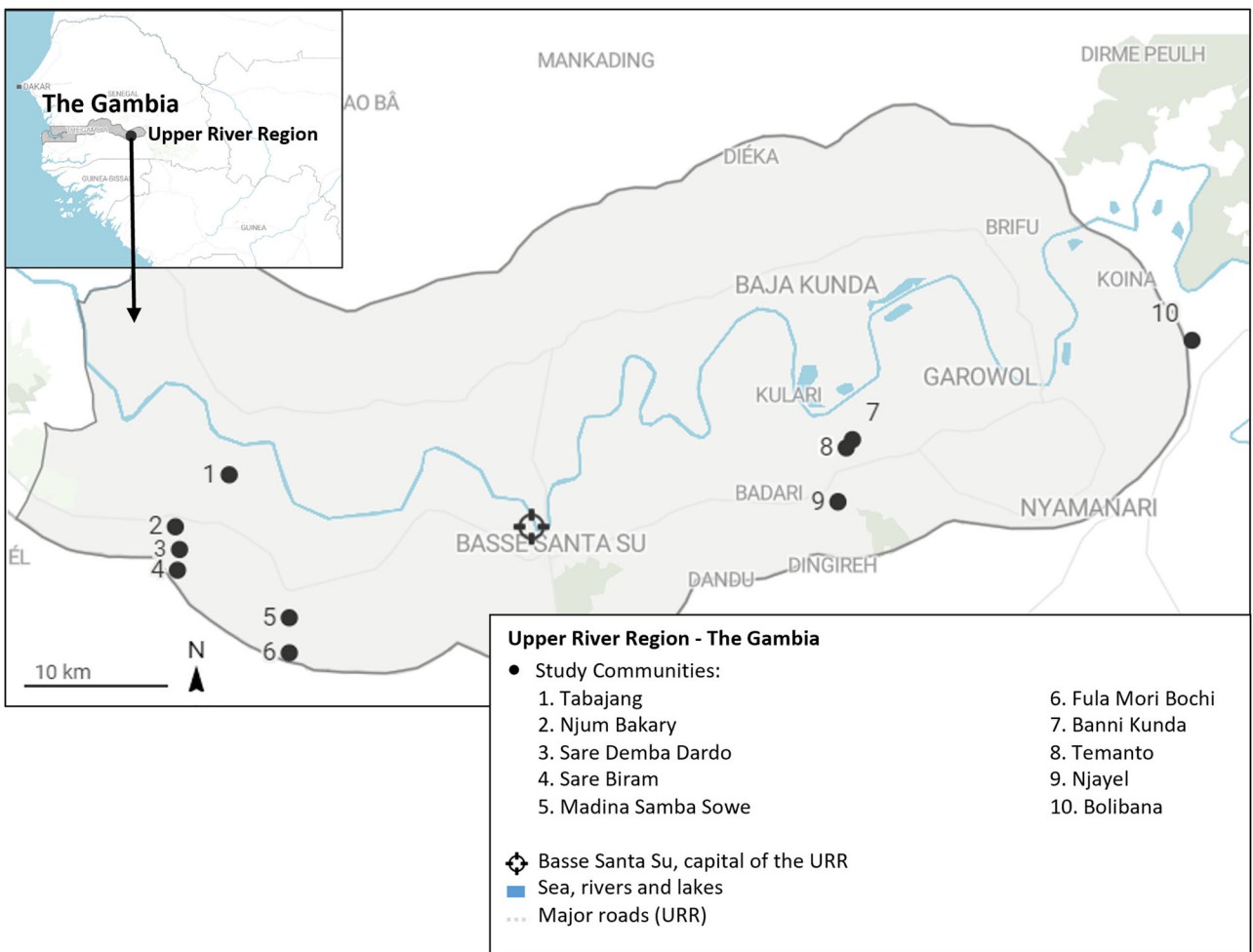

**Fig. 1 | Map of study site.** The locations of the ten study communities are denoted by black markers in the Upper River Region of The Gambia. The inset shows the location of the region within the country.

## Statistical analysis

The impact of SMC status on the incidence of clinical malaria in eligible children was assessed by fitting shared frailty cox models with gamma-distributed random effects specified for person to account for the possibility of repeated clinical episodes within the same person (failures). Where frailty models did not stabilise, robust errors were used instead. To account for ongoing malaria control interventions in each community and background community level risk of malaria, the models pre-specified fixed effects for village ID. Models also included fixed effects for the percentage of all visits to the participant at which an insecticide-treated bed net was used the night before and household ID.

To assess the association between household level coverage of SMC in eligible children and malaria burden in adolescents/adults 10+ years of age, we first defined household SMC coverage in two ways: (i) the percentage of children 0–9 years in each household who received at least 1 round of SMC and (ii) the mean number of rounds of SMC per child. Both definitions were implemented as continuous variables and classed into 'low', 'moderate' and 'high' groups thus: <25%: 25–79%, 80%+ (definition 1) and <0.3, 0.3–2 and >2 rounds/child (definition 2), respectively. The high group cut-offs (80% or >2 rounds) were fixed to be consistent with the optimum minimum level of coverage of child health programmes[32–34]. The low/medium cut-off values were varied for definition 1 in the range 0–55% in 5% increments and for definition 2 in the range 0–0.9 in 0.1 unit increments and implemented in household level models described in Supplementary File 1 STable 3. The models with low groups of <25% and 0.3 had the lowest Akaike and Bayesian Information

Criteria values (AIC and BIC) and were thus selected as the final low-group cut-offs in all household level models.

The impact of household-level coverage of SMC as defined above on clinical malaria episodes in participants 10+ years of age was then assessed by fitting shared frailty Cox models to account for repeated clinical episodes as previously. The impact of household-level coverage of SMC on malaria prevalence in participants 10+ years of age was estimated using logistic regression models with a household-level random effect. We visualised parasite concentration distributions in participants aged 10+ years stratified by household SMC coverage by the generation of kernel density estimate plots and estimated the impacts of household SMC coverage on the prevalence of high-density infections in this group using logistic regression models with a household-level random effect. Equivalent data in children 0–4 and 5–9 years were limited by fewer malaria infections and smaller cell sizes, therefore the additional household-level impacts of SMC on eligible children themselves were assessed in all eligible children aged 0–9 years combined (STables 7, 8 Supplementary File). Where multilevel models failed to converge, we calculated household-level estimates of malaria burden in this age group and regressed these against household-level SMC coverage using a two-stage technique to adjust for individual-level covariates[35,36]. All individual and household-level models of impact of SMC were implemented in Stata version 17.0 (StataCorp, Texas USA).

All household-level models pre-specified fixed effects for village ID, the percentage of all visits to households at which inhabitants used an insecticide-treated bed net the night before, the household level prevalence

**Article**

**Fig. 2 | Flow diagram of study participants and households.** All residents of the study site were eligible to participate in the study and 87% were enroled. Only participants who were present in at least one survey during the 2021 malaria season (July-December) with full address data were included in the final analysis sample.

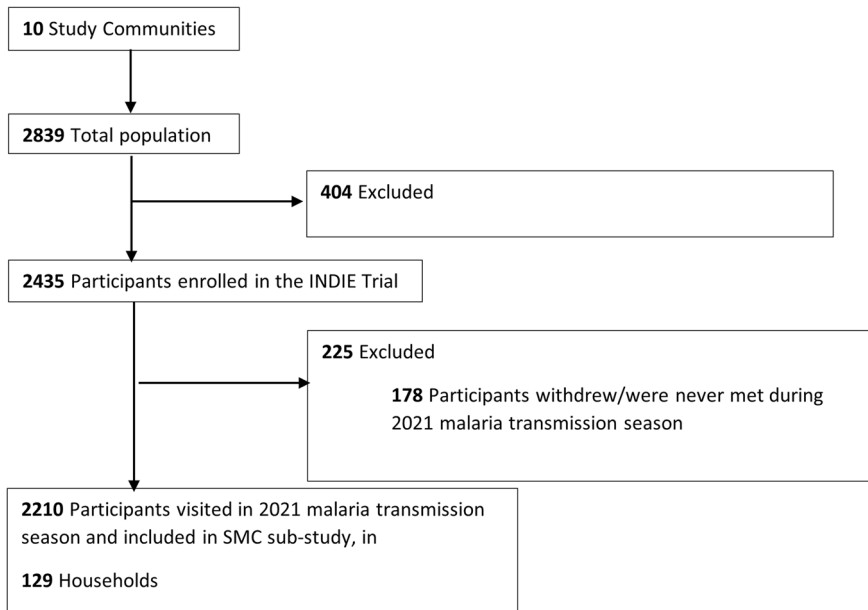

of malaria at a baseline dry season survey in April-May 2021, age in years, the household ratio of children aged 0–9 years to participants aged 10+ years, the total number of household inhabitants and for prevalence models, the week of survey visit.

To assess the effect of spatial clustering of household-level impacts of SMC, an inverse weighted distance matrix was created describing the Euclidean distances (km) between all households in the study site. We calculated the mean incidence rates of clinical malaria per participant aged 10 years and older per enumerated household, and the mean incidence rate in children 0–9 years per household and attached additional features of SMC coverage, insecticide-treated bed net usage, compound size and age composition, and geolocation (latitude and longitude) to each household. Identical datasets were constructed with the overall prevalence of asymptomatic infection per household during the late-season survey for the same age groups. Visual examples of the resulting datasets are shown in Supplementary Fig. 2 (Supplementary File). The distance matrix was used to calculate global Moran indices and z-scores to test for spatial autoregression under the hypothesis that prior regression estimates of the association between household-averaged outcomes for incidence and prevalence of malaria and household SMC coverage were not influenced by estimates in nearby households. Models were adjusted for household level covariates as previously stated - percentage of all visits to households at which inhabitants used an insecticide-treated bed net the night before, the household level prevalence of malaria at a baseline dry season survey in April-May 2021, the household ratio of children aged 0–9 years to participants aged 10+ years and the total number of household inhabitants. Visualisation of spatial data, construction of distance matrices and spatial analysis was conducted using Seaborn, Geoplot, Geopandas and the Esda_Moran libraries in Python (Python Software Foundation, Python Language Reference, version 3.9).

### Ethical approval

This study did not collect any new data or implement additional activities not previously covered by approvals already in place for the INDIE study. Ethical approval for the INDIE study was provided by the Government of The Gambia/The MRC Gambia Joint Ethics Committee, The Gambia, and the Ethics Committee of the LSHTM, UK (reference: 16642).

### Reporting summary

Further information on research design is available in the Nature Portfolio Reporting Summary linked to this article.

## Results

### Characteristics of the study sample

Census data for the site (year 2021) was provided by the Health and Demographic Surveillance System (DSS) team for the Basse District Area. The estimated combined population for the ten study communities was 2839 residents in 144 households, of which 2435 participants in 133 households were enroled into the main INDIE study. The present study included 2210 participants (825 aged 0–9 years and 1,385 participants aged 10+ years) in 129 households who were met during the 2021 malaria transmission season and had complete address data (Fig. 2). Supplementary data 1 summarises the characteristics of study households by the categories of coverage of children with 1+ round SMC in children aged 0–9 years, and Supplementary data 2 summarises the same information by categories of the mean number of SMC rounds per child. Most households had at least 1 malaria case over the season and most had inhabitants in each age category, though a smaller proportion of households within the lowest household SMC coverage categories had inhabitants aged 0–4 years (69.2%) compared to the overall proportion (89.9%). These households also tended to be smaller than the average (10 versus 17 mean inhabitants respectively).

### Impact of SMC status on malaria in eligible children

Dates of SMC rounds were extracted directly from SMC cards for 68.3% of children, and based on caregiver reports for 31.6% of children who had received at least 1 round of SMC. There was no SMC cards available for children who did not receive SMC. Overall, 76.0% of children aged 0–9 years received at least one round of SMC in 2021, and 20.9% received four rounds. Coverage was highest in children aged 0–4 years (90.6% and 31.5% received SMC at least once and all four rounds, respectively) than those aged 5–9 years (64.0% at least one round, 12.1% all four rounds, respectively) (Table 1).

The incidence rates (IR) of clinical malaria over the 2021 season in children eligible for SMC are shown in Table 2 and Kaplan-Meier plots for clinical malaria by SMC coverage are displayed in Supplementary Fig. 1 (Supplementary File). IRs for clinical malaria in children who received at least 1 round of SMC was 1.58 episodes per 100 person-months of follow-up, compared to an IR of 2.89/100 person-months in children who received none. This was highly significant in both unadjusted and adjusted Cox models (adjusted hazard ratio 0.44 95% CI 0.21, 0.93; $p$ value 0.031). The largest reductions were seen in children who received 3–4 rounds of SMC (adjusted hazard ratio 0.38 95% CI 0.16,

**Table 1 | Coverage of Seasonal Malaria chemoprevention in 825 children aged 0–9 years during the 2021 malaria transmission season (MTS), Upper River Region The Gambia**

| SMC round | % children received SMC at each round | | |
|---|---|---|---|
| | 0–9 years (N = 825) | 0–4 years (N = 372) | 5–9 years (N = 453) |
| No SMC received | 24.0% (198) | 9.4% (35) | 36.0 (163) |
| Round 1 (8–19 August 2021) (n) | 52.5% (433) | 73.1% (272) | 35.5% (161) |
| Round 2 (8–17 September 2021) (n) | 59.9% (486) | 72.3% (269) | 47.9% (217) |
| Round 3 (12–16 October 2021) (n) | 50.4% (416) | 59.1% (220) | 43.3% (196) |
| Round 4 (8–12 November 2021) (n) | 42.8% (353) | 57.3% (213) | 30.9% (140) |

| Minimum rounds | % children by minimum number of rounds of SMC received | | |
|---|---|---|---|
| | 0–9 years (N = 825) | 0–4 years (N = 372) | 5–9 years (N = 453) |
| At least 1 | 76.0% (627) | 90.6% (337) | 64.0% (290) |
| At least 2 | 60.1% (496) | 77.2% (287) | 46.1% (209) |
| At least 3 | 47.6% (393) | 62.6% (233) | 35.3% (160) |
| 4 | 20.9% (172) | 31.5% (117) | 12.1% (55) |

0.92; $p$ value 0.031). Reductions were also significant when stratified by age; hazard of clinical infection was very low in children 0–4 years of age who received 1 or more rounds of SMC in the adjusted model. However, associations between SMC and clinical malaria in children aged 5–9 years although consistent with an impact, did not remain significant after adjustment.

STable 4 (Supplementary File) shows malaria prevalence in eligible children by SMC status. Prevalences were low, and lowest in children aged 0–9 years who received 3 or 4 rounds of SMC (2.6%), and higher in those who received only 1–2 rounds (6.5%). Prevalence in children who received no SMC was 4.4%. No impact of SMC coverage on malaria prevalence remained significant in the adjusted models.

### Household level effects of SMC coverage
In Cox models with SMC coverage specified as a continuous exposure, there was an inverse and significant association between clinical malaria in participants 10 years or older and coverage of one or more rounds of SMC in younger children in the same household (HR: 0.99; 95% CI: 0.99, 1.00; $p = 0.039$) and a close to significant inverse association between mean rounds of SMC per child per household and clinical malaria in older participants (HR 0.80; 95% CI: 0.63, 1.02; $p = 0.069$). Predicted trends from these models indicate a 10% decrease in hazard for a 10% rise in the percentage of children who received any SMC, or a 20% decrease in hazard for each additional round of SMC received per eligible child.

In categorical models, we similarly observed reduced incidence of clinical malaria in participants aged 10 years and older with increasing household coverage of SMC (Fig. 3). After adjustment, the reductions were significant or close to significant and of similar magnitude when SMC coverage in the same household was moderate (25–79% of children received 1+ round SMC—adjusted HR: 0.60; 95% CI: 0.34, 1.07; $p = 0.084$), or high (80% or more received 1+ rounds SMC—adjusted HR: 0.49; 95% CI: 0.28, 0.86; $p = 0.014$) compared to households where coverage was low (<25%) (Table 3). Similar significant reductions in clinical malaria were observed in households defined by mean number of SMC rounds per child (Fig. 3 and Table 3).

There was a consistent trend of reduced clinical malaria in children 0–9 years with increasing household coverage of SMC, however, this trend was not significant in the fully adjusted models - except for children who received no SMC residing in households with otherwise high coverage (STable 6 Supplementary File).

Unadjusted continuous mixed effects logistic models estimated significant reductions in malaria prevalence with increasing household SMC coverage in 1056 participants aged 10+ years visited in the late malaria season (unadjusted OR: 0.98; 95% CI: 0.97, 0.10; $p = 0.022$ and OR: 0.64; 95% CI 0.42, 0.97; $p = 0.034$ for SMC coverage defined as percentage of children in household with 1+ SMC rounds or mean SMC rounds per child in household, respectively). These effects did not remain significant after adjustment (adjusted OR: 0.99; 95% CI: 0.98, 1.00; $p = 0.200$ and OR: 0.82: 95% CI: 0.55, 1.23; $p = 0.341$ for SMC as previously). When SMC coverage was categorised as low, medium or high, malaria prevalence in older participants ranged from 6.8% in high coverage households to 18.8% in low coverage households (Fig. 4a, c). In categorical mixed effects models, malaria prevalence in participants aged 10+ years was lowest in households in the two highest coverage groups (moderate and high groups) and both unadjusted and adjusted models suggested a strong decreasing trend in prevalence with increasing SMC coverage (Table 4). The inverse was observed on parasite densities in infected individuals: as household SMC coverage increased, the distribution of parasite densities skews higher (Fig. 4b, d) and the risk of high-density infections increases four-fold or more in households in higher coverage groups (Table 4).

Malaria prevalence was relatively low in 684 children 0–9 years visited in the same survey (all <6.5% STable 7 Supplementary File). We detected no significant associations between household SMC coverage and malaria prevalence in children 0–9 years in adjusted models.

**Table 2 | Incidence and hazard ratios for clinical malaria in 825 children aged 0–9 years by number of rounds of SMC received in the 2021 malaria transmission season, Upper River Region, The Gambia**

| SMC status | Incidence rates per 100 person-months (cases/PM) | | | Unadjusted HR (95% CI) p | | | Fully adjusted HR (95% CI) p | | |
|---|---|---|---|---|---|---|---|---|---|
| | 0–9 years | 0–4 years | 5–9 years | 0–9 years | 0–4 years | 5–9 years | 0–9 years | 0–4 years | 5–9 years |
| No SMC | 2.89 (31/1073.5) | 1.57 (3/191) | 3.17 (28/882.5) | 1 (ref) | 1 (ref) | 1 (ref) | 1 (ref) | 1 (ref) | 1 (ref) |
| 1+ rounds SMC | 1.58 (54/3422) | 1.14 (21/1843) | 2.09 (33/1578.5) | 0.43 (0.27, 0.67) <0.001 | 0.45 (0.13, 1.52) 0.196 | 0.55 (0.33, 0.91) 0.022 | 0.44 (0.21, 0.93) 0.031 | 0.08 (0.01, 0.72) 0.024 | 0.41 (0.13, 1.39) 0.156 |
| 1–2 rounds | 1.88 (24/1274.5) | 1.76 (10/567) | 1.98 (14/708) | 0.56 (0.32, 0.97) 0.037 | 0.84 (0.22, 3.16) 0.791 | 0.53 (0.28, 1.02) 0.056 | 0.50 (0.23, 1.10) 0.084 | 0.13 (0.01, 1.44) 0.097 | 0.29 (0.08, 1.02) 0.053 |
| 3–4 rounds | 1.40 (30/2147) | 0.86 (11/1276) | 2.18 (19/870.5) | 0.36 (0.22, 0.60) <0.001 | 0.31 (0.09, 1.12) 0.074 | 0.56 (0.31, 1.02) 0.060 | 0.38 (0.16, 0.92) 0.031 | 0.05 (0.01, 0.43) 0.006 | 0.60 (0.15, 2.32) 0.460 |

Showing incidence rates as episodes of clinical malaria per 100 person-months (PM) of follow-up. Total cases and total person-months per group in brackets. Adjusted Cox regression models included % LLITN usage over transmission season, village and household ID as fixed effects.

## Spatial impacts of household level SMC coverage

We successfully enumerated 90% (129/144) of all households in the study site. Following regression of mean household IRs or percent prevalence against household level SMC coverage, we assessed the degree of spatial autocorrelation of model residuals using tests of Moran's I. *P* values for all models exceeded 0.05 (0.90–0.99) indicating no spatial clustering of our household level modelling outcomes (Supplementary Fig. 3 Supplementary File).

## Discussion

Empirical evidence for the impact of SMC on malaria transmission is limited but has critical implications for programme scale-up[18]. In this study, we have applied a household-level modelling approach to assess the effects of extended-age SMC on older household inhabitants in a pre-elimination setting in the Gambia. In adjusted models, we observed a significant reduction in the incidence of clinical malaria in eligible children who received SMC compared to those who did not. We observed that older participants ineligible for SMC in households with higher coverage of SMC were significantly less likely to contract clinical malaria compared to households where SMC coverage was low. In a survey of the population late in the malaria transmission season we also observed fewer asymptomatic infections in older participants in households with higher SMC coverage. There was no evidence of spatial clustering of these impacts.

Whilst the incidence of clinical malaria in SMC-eligible children aged 0–9 years decreased with increasing household level SMC coverage and irrespective of their individual level SMC status, we could not reliably demonstrate this constituted an additional benefit of SMC in our adjusted models (STable 6 Supplementary File) neither did we detect an impact of household level coverage of SMC on asymptomatic infections in SMC-eligible children (STable 7 Supplementary File).

The impacts observed in this study broadly align with the results of a 2008-2010 pilot stepped wedge trial which experimentally increased the eligible age range for SMC in Senegal to 10 years[6]—to our knowledge this is the only other source of data on clinical impact following SMC in this age group. The authors reported a decrease in the incidence of clinical malaria in eligible children (69%) and in older age groups (26%). The study did not consider the impacts of lower coverage or impact on asymptomatic infections, and was implemented with excellent research fidelity and high SMC coverage over the three years of implementation, where 84–93% of children received 3 rounds of SMC. A key question for decision-making is whether similar impacts on malaria transmission are achievable in routinely delivered large-scale programmes. Following an era of efficacy and effectiveness trials of SMC initially demonstrating high programme coverage and large effects in study populations, emerging data from routine SMC programmes in West Africa suggests mixed, or reduced impact[8,9,32,37,38]. In our study, coverage of children receiving 3 or more rounds of SMC was 49% overall and only 36% of children 5–9 years; however this was the first season in which the NMCP of the Gambia extended the eligible age for SMC, and the coverage may reflect the initial complexities in operationalising the change. It nonetheless suggests transmission impacts are detectable in the context of programmes with moderate implementation strength.

There is an extensive theoretical basis from both laboratory and field studies which suggest the infectious reservoir for falciparum malaria in sub-Saharan Africa is normally maintained by children up to 15 years of age[12,15,16], and the expected extended benefits of preventative interventions in this age group should be observable in non-eligible adolescents and adults[18]. We justify the use of a household-level models to assess impact of SMC on transmission based on previous research in The Gambia[26] and similar low to moderate endemic settings[27] which indicate malaria transmission is highly localised within household/family groups[39] with negligible contribution from nearby households. This is supported by the null findings from our global Moran tests for spatial autocorrelation in household-level SMC models in the study site. Based on this proposed framework, we hypothesised the mechanism through which herd impacts of SMC arise will be by first reducing infections in the target age group in the same household. We

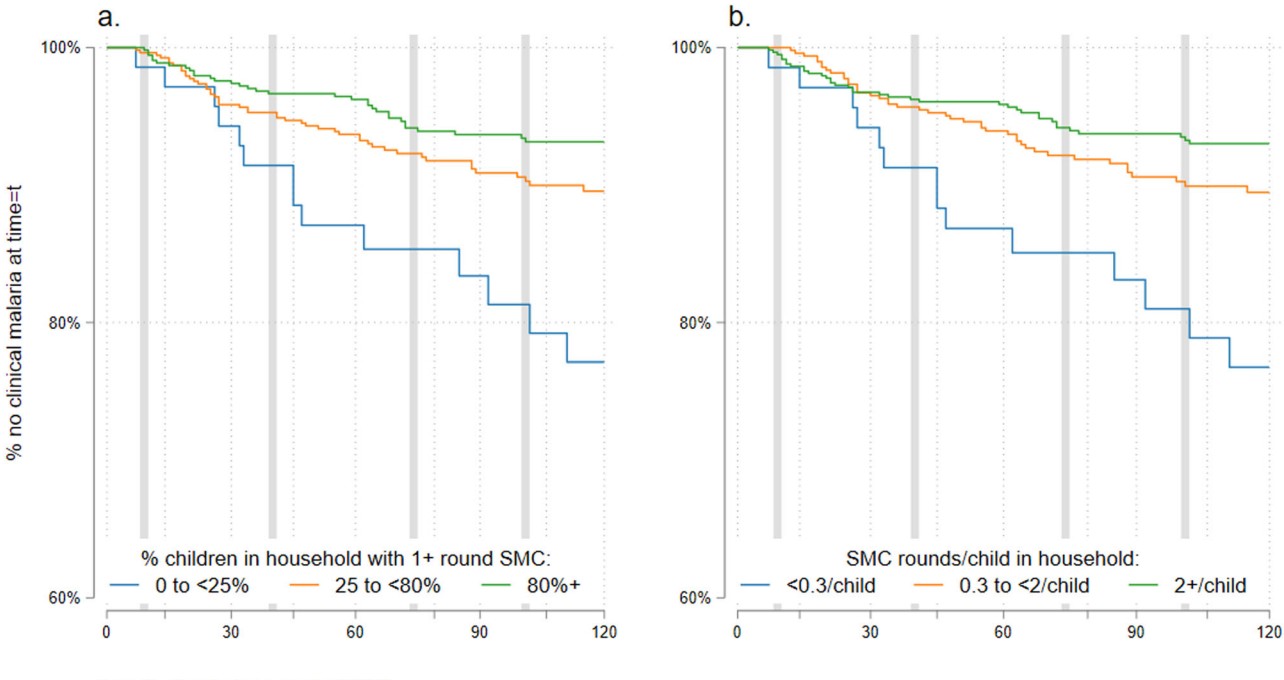

**Fig. 3 | Incidence of clinical malaria by household-level coverage of SMC: Kaplan-Meier survival plots of clinical malaria episodes in 1385 study participants aged 10+ years by SMC coverage in eligible children aged 0–9 years in the same household.** Vertical axes start at '60% no clinical malaria at time = t' for clarity. SMC coverage is defined as (**a**) percentage of eligible children in household with 1+ rounds of SMC, where the blue line is participants in households with <25% coverage, the orange line refers to those in households with 25–<80% coverage and the green line those with 80%+ coverage. **b** Mean number of SMC rounds per eligible child, were blue line is participants in households with <0.3 rounds/child, the orange line refers to those in households with 0.3–<2 rounds per child, and the green line those with 2+ rounds per child. Grey vertical lines indicate the first reported day of each monthly round of SMC.

**Table 3 | Incidence rates and hazard ratios (HR) for clinical malaria in 1385 study participants aged 10+ years stratified by the percentage of children aged 0–9 years in the same household who received one or more rounds of SMC, or the mean number of SMC rounds per child, during the 2021 malaria transmission season in the Upper River Region of The Gambia**

| SMC coverage | Incidence rates per 100 person-months (cases/PM[a]) | Unadjusted HR (95% CI) p | Fully adjusted HR (95% CI) p |
|---|---|---|---|
| % Children in household received at least 1 round SMC | | | |
| 25% | 5.46 (24/439) | 1 (ref) | 1 (ref) |
| 25–75% | 2.51 (90/3585) | 0.52 (0.31, 0.88) 0.015 | 0.60 (0.34, 1.07) 0.084 |
| 80% | 1.49 (51/3434) | 0.29 (0.17, 0.51) < 0.001 | 0.49 (0.28, 0.86) 0.014 |
| SMC Rounds/child in household | | | |
| <0.3 | 5.47 (24/439) | 1 (ref) | 1 (ref) |
| 0.3–<2 | 2.63 (88/3344) | 0.54 (0.32, 0.91) 0.020 | 0.57 (0.33, 0.99) 0.047 |
| >2–4 | 1.44 (53/3675) | 0.29 (0.17, 0.51) < 0.001 | 0.53 (0.28, 1.02) 0.059 |

All models adjusted for village ID, household size, ratio of children to adults, age in years, household ID, baseline (dry season) infection prevalence in household and % of nights household used insecticide treated nets.
[a]Incidence rate per 100 person months of follow-up (total malaria cases/total person months of follow-up).

identified a significant reduction in clinical infections in children 0–9 years, but only decreased asymptomatic infections in children with 3 or more rounds of SMC which did not reach significance (Supplementary File 1 STable 4). Interestingly children who received only 1–2 rounds of SMC benefitted in terms of reduced incidence of clinical episodes but had increased risk of asymptomatic infections compared to children receiving no SMC, though overall prevalences were low (<6.5%). Children who had received only 1–2 SMC round were also more likely to have received SP in the first half of the season (Supplementary File 1 STable 5) perhaps leading to a slightly elevated risk of infection in the following months of rising transmission due to naïve immunity—however, given low prevalence estimates we cannot rule out the potential presence of sampling error. The evidence for the transmission-blocking potential of SP-AQ is conflicting;

in vivo and in vitro studies have indicated exposure to SP may increase commitment of blood stage parasites to sexual differentiation resulting in increased gametocyte concentrations and potential increased transmission potential to feeding mosquitoes[40–42] however detailed analysis of this phenomenon suggests impacts occur for a small number of drug classes under a narrow drug concentration window, and are unlikely to result in a net increase in transmission potential[43]. Pyrimethamine has also shown suppressive activity against oocyte production or successful development of sporozoites in the mosquito[44,45], which may offset increased gametocytogenesis with use of SP. Field studies support a theory of increased gametocytogenesis following SP use or SMC but are also characterised by small sample sizes and incomplete accounting for potential confounding, and alone do not resolve this conflict[46,47]. Whilst gametocyte concentrations were

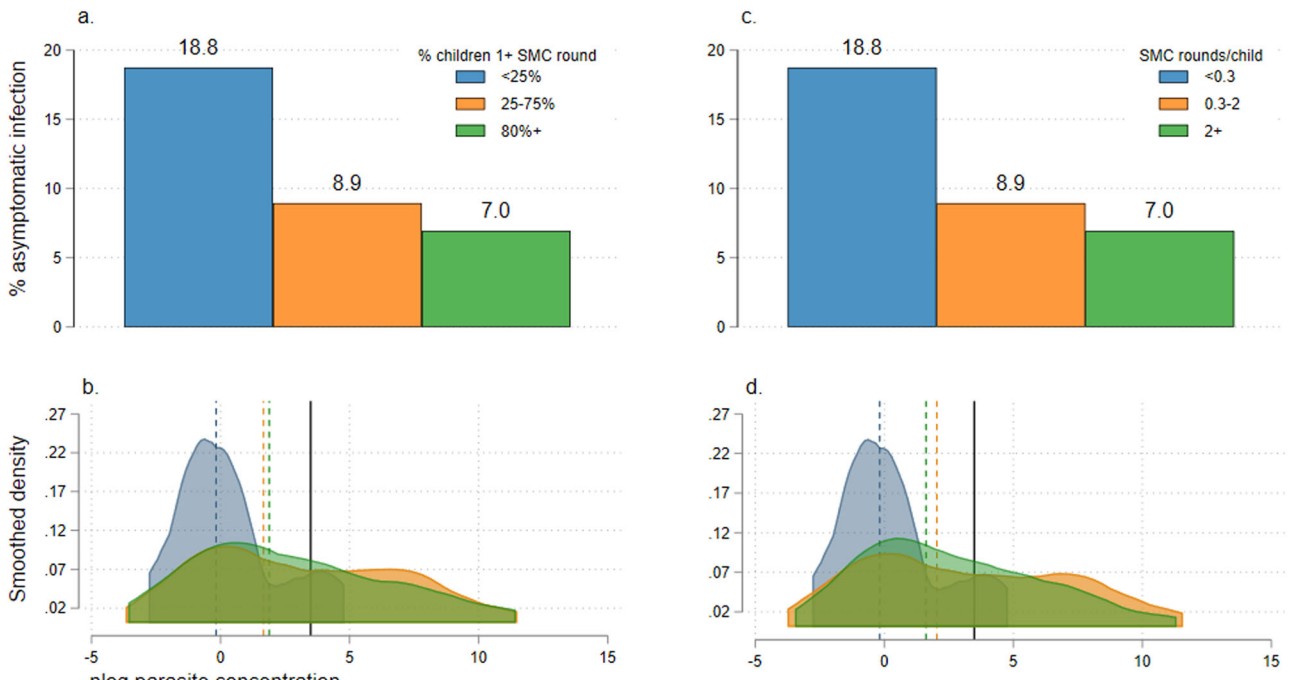

**Fig. 4 | Prevalence and density of asymptomatic infections in 1,056 study participants aged 10+ years surveyed between the 27th Sept and 27th Nov 2021, by household-level coverage of SMC in young children. a** Prevalence of asymptomatic infection in all participants aged 10+ years, by the coverage of SMC in young children in the same household. SMC coverage is defined as the percentage of children 0–9 years who received at least one round of SMC in three groups: <25% (blue bar), 25–75% (orange bar) and 80%+ (green bar). **b** Kernel density plot showing the log (natural) parasite concentrations originally measured in counts per µL in PCR-positive participants aged 10+ years. Plots are stratified by SMC coverage in children 0–9 years in the same household, defined as the percentage of children 0–9 years who received at least one round of SMC in three groups: <25% (blue plot), 25–75% (orange plot) and 80%+ (green plot). Median parasite concentrations for each SMC coverage group are shown by dotted vertical lines (colours to match SMC coverage category); cut-offs for high-density infections (averaged for age groups 10–15 and 16+ years) are shown as solid black lines. **c** Prevalence of asymptomatic infection in all participants aged 10+ years, by the coverage of SMC in young children in the same household. SMC coverage is defined as the mean number of SMC rounds per child also in three groups: <0.3 (blue bar), 0.3–2 (orange bar) and >2 (green bar). **d** Kernel density plot showing the log (natural) parasite concentrations originally measured in counts per µL in PCR-positive participants aged 10+ years. Plots are stratified by SMC coverage in children 0–9 years in the same household, defined as the mean number of SMC rounds per child also in three groups: <0.3 (blue plot), 0.3–2 (orange plot) and >2 (green plot). Median parasite concentrations for each SMC coverage group are shown by dotted vertical lines (colours to match SMC coverage category); cut-offs for high-density infections (averaged for age groups 10–15 and 16+ years) are shown as solid black lines.

**Table 4 | Prevalence of asymptomatic infections in participants 10+ years 27th Sept-29th Nov 2021, by household coverage of SMC in young children**

| Household SMC coverage | Prevalence of asymptomatic infection | | | Prevalence of high parasite density infections[a] | | |
|---|---|---|---|---|---|---|
| | n/N (% prevalence) | Unadjusted RR (95% CI) p | Fully adjusted RR (95% CI) p | n/N (% prevalence) | Unadjusted RR (95% CI) p | Fully adjusted RR (95% CI) p |
| % children in household received at least 1 round SMC | | | | | | |
| 25% | 12/64 (18.8%) | 1 (ref) | 1 (ref) | 3/12 (25.0%) | 1 (ref) | 1 (ref) |
| 25–75% | 45/503 (9.0%) | 0.34 (0.10–1.15) 0.082 | 0.44 (0.20–0.96) 0.040 | 26/45 (57.8%) | 2.31 (0.84–6.36) 0.104 | 4.30 (0.76–25.28) 0.099 |
| 80% | 34/489 (7.0%) | 0.28 (0.08–0.93) 0.039 | 0.52 (0.22–1.18) 0.117 | 22/34 (64.7%) | 2.59 (0.94–7.12) 0.065 | 4.94 (0.80–28.31) 0.086 |
| Household mean number SMC rounds/child | | | | | | |
| <0.3 | 12/64 (18.8%) | 1 (ref) | 1 (ref) | 3/12 (25.0%) | 1 (ref) | 1 (ref) |
| 0.3–<2 | 44/477 (9.2%) | 0.35 (0.10–1.15) 0.084 | 0.40 (0.19–0.93) 0.032 | 26/44 (59.1%) | 2.36 (0.86–6.49) 0.095 | 4.77 (0.91–27.46) 0.080 |
| >2–4 | 35/515 (6.8%) | 0.27 (0.08–0.93) 0.037 | 0.58 (0.24–1.39) 0.220 | 22/35 (62.9%) | 2.51 (0.91–6.92) 0.074 | 5.74 (0.92–35.68) 0.061 |

Adjusted models included village ID, household size, ratio of children to adults, age in years, household ID, baseline (dry season) infection prevalence in household, week of cross-sectional visit and % of nights during season household used insecticide-treated nets when asked as fixed effects.
[a]Participants with natural log parasite concentrations in the top 50% for their age group (0–4, 5–9, 10–15, 16+ years) were categorised as high-density infections.

not available for participants in our study, we did observe a trend of increased parasite concentrations in children with SMC and in older participants in households with higher SMC coverage known to be positively associated with gametocyte density[12], however, infection rates were too small to confirm this relationship. Other studies have shown that densities of both asexual and gametocyte stages are nonetheless considerably lower in asymptomatic infections compared to clinical disease[48,49]. This current body of evidence highlights the mechanisms of action of SP-AQ SMC are complex and not yet fully elucidated; the net product of a trade-off between reduced clinical infection, increased asymptomatic parasitaemia and

gametocytaemia and potential suppressive effects on mosquito lifecycle stages may be a reduction in overall transmission potential.

An assessment of the transmission-reducing effects of SMC is most useful for policy if based on results of routinely delivered programmes. This setting better predicts real-life impacts under sub-optimal or heterogenous SMC coverage and in the presence of other nationally implemented malaria interventions. High intervention coverage and strict eligibility criteria for participants and clusters observed in trial settings often do not transfer with high fidelity or effectiveness for this reason. Nonetheless, a limitation of our observational approach is determining the appropriate techniques to account for other exogenous predictors of transmission that may also differ by SMC coverage, not all of which can be measured. We included a priori fixed effects in statistical models to address this. Village ID was used as a proxy for underlying transmission potential for a community, clustering by ethnicity and in the context of the wider INDIE study also allowed us to adjust for ongoing interventions that differed by community. We also included household-level baseline malaria infection prevalence and indicators of individual or household level bed net usage - variation in treated bed net usage can predict mosquito biting rates, local intensity of malaria transmission and/or potential to adhere to other protective behaviours and may correlate with higher acceptability and adherence to SMC programmes. We examined other characteristics of households by SMC coverage group (Supplementary Data 1, 2) to identify potentially confounding characteristics. All else being equal, the household impact of SMC may be influenced by the number of children relative to older inhabitants and/or total household size. These parameters differed by SMC coverage level in this population. We, therefore, addressed these sources of potential confounding by presenting both unadjusted models and models adjusted for the above features with correction of errors to account for clustering at person or household level. We excluded some households from our spatial analysis sample - which detected no geographical clustering of effects of household-level models of SMC - however the vast majority of households (90%, 129/144) across the site were successfully enumerated and the missing (15 households) were spread amongst all communities (Supplementary Fig. 2). Given *P* values for Moran tests were close to 1 it is unlikely data from the 15 excluded households would significantly modify these. As a robustness test, we defined coverage of SMC in two ways, as the percentage of children receiving any SMC, and the mean number of SMC rounds per child with both definitions showing similar patterns of impact. There may be some misclassification of SMC status as the NMCP did not have a policy of directly observed treatment for SMC at the time, and the status for a third of children was based only on caregiver recall. However given the short length of the transmission season and the short length of time between our surveys and the SMC rounds themselves, we expect the levels of misclassification to be relatively low. Our study did not allow us to separate differential herd impacts of SMC coverage by age (0–4 years and 5–9 years), highlighting a potential focus for further research. We also did not perform a detailed assessment of additional herd impacts of household SMC coverage on eligible children irrespective of their own status. In both cases malaria outcomes were relatively less common in this group and in the case of the second initially significant positive effects did not survive adjustment for key confounders. Given wide confidence intervals (STable 6) including the potential for both very large reductions or increases in hazard of clinical malaria, it is likely that this sub-analysis is underpowered. Therefore the possible impacts in this age group therefore remain to be established in future studies. Whilst a cost analysis was outside the scope of this study, estimates from the Senegal study where SMC was extended to the same age range suggest the programme can be implemented with an average delivery cost of $0.50 per child per month[6], however, additional context-specific data of extended programmes from other sites would be useful for policy. Our study was conducted in a moderate/low transmission setting—future studies in higher burden settings will add to this growing evidence base.

In conclusion, we show in the context of a routine extended-age chemoprevention programme with SP-AQ that SMC is associated with a significant reduction in clinical malaria in direct recipients and with a significant reduction in clinical and asymptomatic infection in older household inhabitants. Our data demonstrate these effects are robust to adjustment, are household-specific with little negligible from nearby households. The results support findings of previous laboratory and simulation studies of the importance of children to the infectious reservoir for malaria, addressing a critical evidence gap in demonstrating herd effects in practice. They point to important additional benefits of SMC in reducing overall malaria transmission, highlighting such programmes are likely to be more cost-effective than currently estimated.

## Data availability
Study data (for Tables and in-line results) will be made publicly accessible after the publication of the impact paper for the INDIE 1b project. Datasets will be available via the LSHTM Data Compass data repository (https://datacompass.lshtm.ac.uk/), or from the corresponding authors upon reasonable request. The source data behind the graphs in the figures can be found in Supplementary Data 3.

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

## Acknowledgements

We would like to acknowledge the contribution of all the fieldworkers and participants of the 10 Upper River Region communities who provided data for this study; Isabel Byrne for advice on the spatial analysis approach, and Professor Paul Milligan for information on the status of extended-age SMC pilots in West Africa. This study was funded via a grant to C.D. and T.B. from the Bill and Melinda Gates Foundation (grant ref: OPP1173572). The Foundation did not provide any input to any aspect of the design of the study or development of this manuscript.

## Author contributions

S.S., B.C., U.D.A., T.B., A.E., M.M. and C.D.—responsible for study concept and design. B.C., M.M., A.E., T.B., C.D.—responsible for fieldwork coordination. A.N., S.S., M.M.—responsible for data management. H.M.S.—responsible for entomological surveillance data collection. B.E., M.O.N.—responsible for laboratory assays. S.S., J.B.—responsible for statistical analysis. S.S. wrote the first draft of the paper and all authors reviewed and approved the manuscript.

## Competing interests

The authors declare no competing interests.
