## [Peer Review File · Communications Medicine]

Reviewers' comments:

Reviewer #1 (Remarks to the Author):

The authors presented an analysis on the implementation of seasonal malaria chemoprevention (SMC) in higher age groups based on the updated 2022 World Health Organization (WHO) recommendations. It is of relevance to people in this field and can be used as a base from which additional modelling can be built on for areas of high malaria endemicity in the future. The methodology adopted in the manuscript accounts for confounding factors that could impact the results of the study, showing foresight of potential externalities in implementing SMC in the study area. The fact that the study looks at the implications of the different round uptakes of SMC and their overall importance on malaria cases is commendable for recommendations to policy makers.

While the impact of implementing SMC in the age group 0-9 years was adequately explored, the manuscript can benefit from presenting a minor cost analysis associated with additional SMC for the age group 5-9 years as cost is important when deciding on which interventions to adopt for a certain community or area.

The authors alluded to the fact that they could not reliably demonstrate the additional benefit in the decrease in incidence of clinical malaria in children aged 0-9 years with increasing household level SMC coverage and irrespective of their individual level SMC status in their adjusted models, it would be of additional importance to provide a possible or plausible explanation to this result.

For additional impact the authors may wish to identify opportunities for gaps that can be addressed in future studies.

Reviewer #2 (Remarks to the Author):

Minor comments (mostly proofreading)

Line 42: 'demonstrate that'

Line 45: 'assessed these effects with ...'

Line 67: Reference 10 is a qualitative paper about operational aspects, not a cost-benefit analysis.

Line 127: the quoted HR is very close to 1.0, which would correspond to no effect. Could there be a typo or problem of units here? It is hard to see how such a small effect size could be statistically significant if the explanatory variable is coded 0, 1. There is a similar question about the OR on line 143 where the CI looks inconsistent with the quoted OR.

Line 249: I think this should read 'Empirical evidence'. The fact that there is a clear mechanism

and that we know that SMC reduces gametocytaemia is surely evidence of a sort.

Line 317: 'included in statistical ...'

Line 339: 'nor did it allow' instead of 'nor for a'

Figure 3. What are the units of parasite concentration that have been log transformed?

Stable 1: what is the figure in parentheses (the standard deviation)? What was the total number of trap-nights? Does this need its own table as the numbers are quoted in the main text?

Stable 2: Are these the means of the daily maximum temperatures or the maxima of the daily maxima?

SFigure 1: what is the point of the left hand column of figure panels? The same information, and more, appears in the right hand column. There is no explanation in the legend of the difference between the two columns.

Response to reviewers comments

11th November 2023

Referee #1: malaria, infectious disease modelling and statistics, Africa

1. The authors presented an analysis on the implementation of seasonal malaria chemoprevention (SMC) in higher age groups based on the updated 2022 World Health Organization (WHO) recommendations. It is of relevance to people in this field and can be used as a base from which additional modelling can be built on for areas of high malaria endemicity in the future. The methodology adopted in the manuscript accounts for confounding factors that could impact the results of the study, showing foresight of potential externalities in implementing SMC in the study area. The fact that the study looks at the implications of the different round uptakes of SMC and their overall importance on malaria cases is commendable for recommendations to policy makers.

2. While the impact of implementing SMC in the age group 0-9 years was adequately explored, the manuscript can benefit from presenting a minor cost analysis associated with additional SMC for the age group 5-9 years as cost is important when deciding on which interventions to adopt for a certain community or area.

Response: Estimating costs of the extended SMC programme would indeed provide valuable information. Such work is however outside of the scope of this paper - the relevant data to conduct this analysis unfortunately will not be obtainable from the national malaria control programme in a timely manner. We believe this would be an important aspect of future work on this topic. We have added a sentence recommending cost data is collected in future studies of extended age SMC:

Page 23 line 352: "Whilst a cost-analysis was outside the scope of this study, estimates from the Senegal study where SMC was extended to the same age range suggests the programme can be implemented with an average delivery cost of \$0.50 per child per month, however additional context-specific data of extended programmes from other sites would be useful for policy."

3. The authors alluded to the fact that they could not reliably demonstrate the additional benefit in the decrease in incidence of clinical malaria in children aged 0-9 years with increasing household level SMC coverage and irrespective of their individual level SMC status in their adjusted models, it would be of additional importance to provide a possible or plausible explanation to this result.

Response: Thank you, we have now clarified this. The impact of household SMC coverage on children 0-9yrs is reported in Supplement STable 7. In all eligible children the p values suggested no impact, however hazard ratios varied considerably with confidence intervals in the approximate range between a 75% reduction to a 300%+ increase in hazard. In the subgroup of eligible children who themselves did not receive SMC, we again observed no significant impact of household SMC coverage though, similarly some estimates were implausible and all confidence intervals included the potential for large reductions or increases in hazard. Further splitting the data by age 0-4 and 5-9 yrs resulted in model convergence issues. Our best explanation for this is a lack of sufficient

power for this sub-analysis, and we recommend it is repeated on a larger sample of children. We have updated the Discussion section to clarify this:

Page 23 line 348: "...Given confidence intervals (STable 7) including the potential for both very large reductions or increases in hazard of clinical malaria it is likely that this sub-analysis is underpowered. Therefore the possible impacts in this age group therefore remains to be established in future studies."

4. For additional impact the authors may wish to identify opportunities for gaps that can be addressed in future studies.

Response: Noted. In addition to identifying opportunities for future studies of herd effects in unprotected but SMC-eligible children described in the previous response, we have better highlighted the Discussion section detailing some useful evidence gaps:

Page 23 line 344: ..."Our study did not allow us to separate differential herd impacts of SMC coverage by age (0-4 years and 5-9 years), highlighting a potential focus for further research. We did not perform a detailed assessment of additional herd impacts of household SMC coverage on eligible children irrespective of their own status. In the case of the second, malaria outcomes were relatively less common in this group and initially significant positive effects did not survive adjustment for key confounders. Given wide confidence intervals (STable 7) including the potential for both very large reductions or increases in hazard of clinical malaria it is likely that this sub-analysis is underpowered – therefore the possible impacts in this age group therefore remains to be established in future studies. Whilst a cost-analysis was outside the scope of this study, estimates from the Senegal study where SMC was extended to the same age range suggests the programme can be implemented with an average delivery cost of \$0.50 per child per month, however additional context-specific data of extended programmes from other sites would be useful for policy. Our study was conducted in a moderate/low transmission setting – future studies in higher burden settings will add to this growing evidence base."

Referee #2: malaria epidemiology, simulation modelling

Minor comments (mostly proofreading)

Response: Thanks for these updates – we have accepted many of these and corrected the manuscript accordingly.

1. Line 42: 'demonstrate that'

Response: Corrected

Page 2 line 42: “...Upper River Region to demonstrate **that** the hazard...”

2. Line 45: 'assessed these effects with ...'

Response: No change; we assessed whether the effects showed evidence of the phenomenon termed spatial autoregression, the current wording is correct.

3. Line 67: Reference 10 is a qualitative paper about operational aspects, not a cost-benefit analysis.

Response: Noted, our summary may be misunderstood as a formal cost benefit analysis – the authors of the Moukénét et al 2022 paper included a discussion of the resources and costs required to extend the programme and deemed these a barrier. We have updated the wording to make this clearer:

Page 3 line 66: “...and maintenance of effective coverage in the current 0-4 age group have led to reservations about **the feasibility and resources required to implement** an age increase”

4. Line 127: the quoted HR is very close to 1.0, which would correspond to no effect. Could there be a typo or problem of units here? It is hard to see how such a small effect size could be statistically significant if the explanatory variable is coded 0, 1. There is a similar question about the OR on line 143 where the CI looks inconsistent with the quoted OR.

Response: We have updated this with improved wording at the beginning of this paragraph, which now clarifies the first results are based on continuous exposure variables. The ratios are therefore a unit increase in hazard for a 1 unit increase in coverage indicator (i. percentage of children with any SMC or ii. SMC rounds/child). Note we also provide a different perspective on the interpretation of the hazard ratios in line 130:

Page 6 line 126: “In **Cox models with SMC coverage specified as a continuous exposure...**”

Page 6 line 130: “Predicted trends from these models indicate a 10% decrease in hazard for a 10% rise in the percentage of children who received any SMC, or a 20% decrease in hazard for each additional round of SMC received per eligible child.”

5. Line 249: I think this should read 'Empirical evidence'. The fact that there is a clear mechanism and that we know that SMC reduces gametocytaemia is surely evidence of a sort.

Response: Noted and changed

Page 20 line 255: “**Empirical** evidence for the impact of SMC on malaria transmission is limited...”

6. Line 317: ‘included in statistical ...’

Response: Noted and changed

Page 22 line 323: “We included a priori fixed effects in statistical models”

7. Line 339: ‘nor did it allow’ instead of ‘nor for a’

Response. In response to a previous reviewer comment this sentence has changed thus:

Page 23 line 344: ...”Our study did not allow us to separate differential herd impacts of SMC coverage by age (0-4 years and 5-9 years), **highlighting a potential focus for further research.** **We did not** perform a detailed assessment of additional herd impacts of household SMC coverage on eligible children irrespective of their own status.

8. Figure 3. What are the units of parasite concentration that have been log transformed?

Response: We have provided this information in the Figure legend:

Page 17 line 234: **b) Log (natural) parasite concentrations originally measured in counts per μL**

9. Stable 1: what is the figure in parentheses (the standard deviation)? What was the total number of trap-nights? Does this need its own table as the numbers are quoted in the main text?

Response: The value in parentheses is the standard deviation. We have retained the preamble and table to include details on min-max values and trap nights:

Supplement page 1 line 7: “Each **month over this period nightly catches took place over three days** in a random selection of 6 households in each village, using Centre for Disease Control (CDC) light traps.”

Supplement page 1 Stable 1: “Mean nightly biting rate (**standard deviation**)*” [column two]; “**Total trap-nights 829**” [column three]

10. Stable 2: Are these the means of the daily maximum temperatures or the maxima of the daily maxima?

Response: These are the maximum temperatures over the entire month, and are not averaged. We’ve made a change to clarify this:

Supplement page 1 line 10: “Stable 2. **Maximum** temperature and **mean** rainfall data from Basse Meteorological Station 2019-2021”

11. SFigure 1: what is the point of the left hand column of figure panels? The same information, and more, appears in the right hand column. There is no explanation in the legend of the difference between the two columns.

Response: Thank you for spotting this omission. The left hand column shows overall impact of ‘any SMC’ (left column) versus ‘no SMC’, and the right hand column provides a further breakdown of the ‘any SMC’ group – we believe both results are informative. A legend has been added:

Supplement page 7 line 56: “Left hand column: SMC coverage as binary categorisation (no SMC/any SMC). Right hand column: No SMC, Any SMC category further divided into 1-2 rounds and 3-4 rounds of SMC”.

REVIEWERS' COMMENTS:

Reviewer #1 (Remarks to the Author):

The authors have adequately addressed the reviewers' concerns

Reviewer #2 (Remarks to the Author):

The authors have adequately addressed the criticisms of the previous version.